# Impact of Postharvest Handling on Preharvest Latent Infections Caused by *Monilinia* spp. in Nectarines

**DOI:** 10.3390/jof6040266

**Published:** 2020-11-04

**Authors:** Carlos Garcia-Benitez, Carla Casals, Josep Usall, Ismael Sánchez-Ramos, Paloma Melgarejo, Antonieta De Cal

**Affiliations:** 1Department of Plant Protection, Instituto Nacional de Investigación y Tecnología Agraria y Alimentaria (INIA), Carretera de La Coruña km 7, 28040 Madrid, Spain; carlosgarciabenitez@gmail.com (C.G.-B.); ismael@inia.es (I.S.-R.); 2IRTA, Postcollita-XaRTA, Parc Científic i Tecnològic Agroalimentari de Lleida, Parc de Gardeny, Edifici Fruitcentre, 25003 Lleida, Spain; Carla.Casals@irta.cat (C.C.); Josep.usall@irta.cat (J.U.); 3Direccion General de Producciones y Mercados Agrarios, MAPA, 28010 Madrid, Spain; pmelgarejo@mapa.es

**Keywords:** brown rot, water dumping, cold storage, degree-days, Gompertz model, stone fruit

## Abstract

Latent infections caused by *Monilinia* spp. in nectarines cause great economic losses since they are not detected and rejected at harvest and can appear at any time post-harvest, even at the consumer’s home. The effect of a pre-cooling chamber, water dump operation, and cold-storage chamber on the activation and/or development of preharvest latent infections caused by *Monilinia* spp. on nectarines were studied under different postharvest conditions: (a) cold storage for 0, 1, or 3 d at 4 °C at either 75% relative humidity (RH) or 100% RH before water dumping, (b) water dumping for 10 minutes at 15 °C, and (c) cold storage for 0, 3, or 10 d at 4 °C at either 75% RH or 100% RH after water dumping. These storage conditions were transformed to fungal physiological time. For visualization of the latent infections caused by *Monilinia* spp., the nectarines were placed in sterile paper bags and frozen at −20 °C for 48 h in order to damage the epidermis. To compare different handling scenarios, the incidence of latent infection was modelled for physiological time description by a modified Gompertz model. The activation and/or development of preharvest natural latent infections caused by *Monilinia* spp. at postharvest was mainly related to temperature and incubation time at postharvest. Storing nectarines with any postharvest handling less than 11 days at 4 °C avoids brown rot symptoms and reduced the activation and/or development of pre-harvest latent infections caused by *Monilinia* spp., while more cold days caused the exponential phase of latent infection activation and/or development. The Gompertz model employed could be used for predicting the activation and/or development of latent infection caused by *Monilinia* spp. at postharvest conditions and looks at the postharvest life. To our knowledge, this is the first time that the effects of post-harvest handling on latent infections in fruit have been studied.

## 1. Introduction

Brown rot is an economically important fungal disease of peaches (*Prunus persica* (L.) Batsch), nectarines (*P. persica* var. *nucipersica* (Suckow) C. K. Schneid), and other stone fruit species. The main causal agents of brown rot in Spain are *Monilinia fructicola* (G Winter) and *M. laxa* (Aderhold & Ruhland) [1]. Although, *M. fructicola* has started to displace *M. laxa* as the major species causing brown rot in the Ebro Valley, Spain [1]. However, this shift was not associated with an increased disease incidence [2].

For fruit infection to occur, *Monilinia* conidia must first adhere to the fruit’s surface before penetrating the surface through natural openings and/or injured areas [3,4]. Germ tube and appressorial formation by *Monilinia* spp. occur during the first stage of the infection process and this infection process is influenced by the ambient temperature, the relative humidity (RH), and the stage of fruit maturity [5]. After penetration, *M. fructicola* and probably *M. laxa* first colonize the stomata and then cause collapse of the epidermal and sub-epidermal cells [6].

However, these infections could also remain latent in immature fruit when fruit physiology and climatic conditions are unfavorable and could persist as latent infections during crop season until the conditions become conducive to disease expression [7,8,9]. Latent infections have been described as a dynamic equilibrium between the host, the pathogen, and the environment without visible symptoms of the disease or signs of the pathogen [10], unlike quiescent infections that produce visible symptoms [6]. Both *M. fructicola* and *M. laxa* have been isolated from peaches, nectarines, and plums with latent infections caused by *Monilinia* spp. [2,10,11,12,13,14,15]. The incidence of latent infections caused by *Monilinia* spp. in harvested fruit usually ranges from 0% to 30%, but it may be as high as 50% [2,10,11,16,17]. Nectarine tissues at harvest with latent *M. fructicola* infections are characterized by the presence of intercellular hyphae at the subcuticular level, which do not remain completely dormant but slowly colonize the tissue [6]. Although a pathogen has a low metabolic level in a latent infection, pathogenicity factors may be activated to end the period of latent infection when conditions for disease development become conducive. The incidence of brown rot in harvested fruit has been positively correlated to the incidence of latent infections caused by *M. fructicola* or *M. laxa* in the field [9,10].

The incidence of brown rot in harvested fruit is the sum of the number of (a) active *Monilinia* infections in harvested fruit, (b) latent infections caused by *Monilinia* spp. in the field, and (c) new *Monilinia* infections that occur during postharvest processing [2,18]. *Monilinia* conidia can be transferred to uninfected harvested fruit by air movement in storage areas or by direct contact with other fruit [18]. However, one can assume that the occurrence of new infections in harvested fruit is minimal because the handling of harvested fruit usually lasts less than one month and is done at low temperatures in order to maximize the fruit’s shelf life and minimize the pathogen’s growth [18]. The harvest season is one of the best times for latent infections to become active, since the fruit is at the most susceptible time and environmental conditions are also favorable (Byrde and Willetts, 1977; García-Benítez et al., 2016), although not all brown rot latent infections are activated by harvest time.

The quality of postharvest fruit depends on careful harvesting, good production practices, and suitable postharvest handling systems. Stone fruit are characterized by about a fortnight of postharvest life, passing quickly from ideal ripeness to over-mature fruit, depending in part on temperature and handling practices (Kader and Mitchell, 1989). The most common methods used in Spain to cool stone fruit is storage in a pre-cooling chamber at 4 or 0 °C in order to slow down metabolism and reduce fruit deterioration [18]. After cooling, fruit are sorted, starting with the water dump operation (with or without disinfectant), where water helps to avoid blows during fruit box overturning. Then, fruit are transported from the tank to the lines with a conveyor belt and rotten fruit are discarded manually [18]. During sorting operations, fruit are selected in order to eliminate defects and sometimes to select fruit with a range of colours, sizes, and shapes. The use of a forced air cooling tunnel (cold storage chamber) is common practice before shipping. Finally, fruit are usually kept at an ambient temperature (25 °C) in the destination market [18].

Against this background, we undertook an investigation to study the effect of a pre-cooling chamber, water dump operation, and cold-storage chamber on the activation and/or development of field latent infections caused by *Monilinia* spp. in harvested fruit at postharvest conditions. Specifically, we investigated the effect of water dumping, the RH levels, and the duration of cold storage in the pre-cooling and shipping areas on the activation and/or development of latent infections caused by *Monilinia* spp. in nectarines. Then, the relationship between the latent infection developmental rate and the temperature regime of storage, expressed as degree days, was fitted by means of a modified Gompertz nonlinear model and their predictions were compared with empirical developmental data recorded on two nectarine cultivars.

## 2. Materials and Methods

To study the effect of different postharvest handling stages on the activation and/or development of field latent infections caused by *Monilinia* spp. in harvested fruit, two completed experiments using two different cultivars of harvested nectarines (“Red Jim” and “Alba Red”) at phenological growth stage BBCH 87 (BBCH: general scale from Biologische Bundesanstalt, Bundessortenamt, and Chemische Industrie, Germany [19]) and without visible signs of brown rot were carried out. Commercial maturity at harvest was defined by the size and colour of the fruit. Immediately after harvest, three lots of 50 kg (≈670 fruit) of each variety were pre-stored at 0 °C until they were sent to the laboratory by 24 h courier.

The natural incidence of latent infections caused by *Monilinia* spp. in each cultivar before their postharvest handling was determined for each experiment by placing 30 frozen surface disinfected nectarines [11,16] in a humidity chamber lined with sterilized moist filter paper and incubated at 25 °C under fluorescent lighting (100 μE m^−2^ s^−1^ with a 16 h photoperiod) for seven days at 100% RH. At the end of the incubation period, the incidence of latent infections caused by *Monilinia* spp. was expressed as a percentage of the total number of nectarines. The incidence of brown rot in harvested fruit before their handling was also determined for each experiment by placing 30 nectarines in a humidity chamber lined with sterilized moist filter paper and incubating them at 25 °C under fluorescent lighting (100 μE m^−2^ s^−1^ with a 16 h photoperiod) for seven days at 100% RH. At the end of the incubation period, the incidence of brown rot was expressed as a percentage of the total number of nectarines. *Monilinia* spp. were isolated and identified morphologically [20] and when in doubt, by polymerase chain reaction (PCR) [21] from all visible and latent infections, which developed during the investigation. Data on the incidence of brown rot and latent infection at harvest time were analysed by chi-squared tests with Yates’ correction for 2-by-2 frequency tables [22] using Statgraphics Centurion XVI for Windows, Version 16.1.03 (StatPoint Technologies, Inc., Herndon, VA, USA).

To know the effect of postharvest handling on activation and/or the development of latent infection, 540 brown rot asymptomatic nectarines from each cultivar were surface disinfected by an initial immersion in 1% sodium hypochlorite for 5 min, followed by an immersion in 70% ethanol for 1 min and two 1 min washes in sterile distilled water in each postharvest handling experiment. Nectarines were disinfected to avoid surface infections and preharvest fungicides residues on harvested fruit. The nectarines were dried by placing them in a laminar flow hood for 2 h [23]. For visualization of the latent infections caused by *Monilinia* spp., the nectarines were placed in sterile paper bags and frozen at −20 °C for 48 h in order to damage the epidermis [17]. After freezing, the nectarines were transferred to sterile Petri dishes on sterile trays (10 nectarines/tray), which were lined with moist sterilized filter paper. Three trays were randomly selected for each of the eight different postharvest treatments, which simulated (a) cold storage for 0, 1, or 3 d at 4 °C at either 75% RH or 100% RH before water dumping, (b) water dumping for 10 minutes at 15 °C, and (c) cold storage for 0, 3, or 10 d at 4 °C at either 75% RH or 100% RH after water dumping (Table 1). In order to simulate the commercial and market period, the nectarines were incubated at 25 °C under fluorescent lighting (100 μE m^−2^ s^−1^ with a 16 h photoperiod) for at least 7 additional days after treatment, until the total incubation period of each treatment was 20 d. The control group comprised two groups of nectarines, each of which consisted of three trays of 10 nectarines and were continuously incubated at 25 °C under fluorescent lighting (100 μE m^−2^ s^−1^ with a 16 h photoperiod) for 20 d at either 75% RH or 100% RH in order to determine the baseline incidence of latent infections caused by *Monilinia* spp. The number of nectarines with latent infections caused by *Monilinia* spp. was recorded daily by counting the number of nectarines with visible signs of brown rot. The infected nectarines were removed in order to prevent secondary infections during their incubation at 25 °C. The RH of the incubation chamber was checked using a temperature/humidity data logger (OM-73 Omega, Stamford, CT, USA).

The incidence of latent infection was calculated for each treatment and the sampling date and the latent infection curve (LIC) were obtained for each one. The area under the latent infection curve (AULIC), in units of percent-days, was calculated by trapezoidal integration [24]. AULIC data were analyzed by analysis of variance [25] using Statgraphics Centurion XVI for Windows, Version 16.1.03 (StatPoint Technologies, Inc., Herndon, VA, USA). When F-test was significant at *p* ≤ 0.05, the means were compared using the Student-Newman-Keuls multiple range test.

In order to study the effect of handling activities on latent infection, incidence at standardized storage conditions for each treatment were transformed to fungal physiological time (Degree-days above 0 °C; DD) (Equation (1)). Degree-days above 0 °C was cumulative daily thermal time, considering that 0 °C is set as the temperature at which *Monilinia* spp. cannot grow [26].

DD = Σ [(Trc × trc) + (Twd × twd) + (Tec × tec)]/24,
(1)
where DD: Degree-days above 0 °C.Trc: Temperature on the pre-cooling chamber (°C).trc: time on the pre-cooling chamber (h).Twd: Temperature on water dump (°C).twd: time on water dump (h).Tec: Temperature on the cold-storage chamber (°C).tec: time on the cold-storage chamber (h).

To compare different handling scenarios and to standardize the impact of temperature on physiological time in each treatment, the incidence of latent infection was modelled for physiological time description. The percentages of latent infections recorded in each treatment for DD were fitted to the following modified Gompertz equation [27] with Tablecurve 2D 5.01 by means of iterative least squares estimation using the Levenberg–Marquardt algorithm:
Y = Ae (^−e^[(μm 2.718)/A (λ − t) + 1]),
(2)
where y: percentage of latent infection.t: degree-days above 0 °C (DD).A: maximum latent infection reached.μ_m_: maximum fungal growth rate.λ: lag phase duration before the beginning of latent infection growth.

Maximum latent infection, maximum fungal growth rate, and lag phase duration were parameters to be estimated from the data. Furthermore, the incidence of latent infection in the different treatments was compared using a linear mixed effects model [28,29]. Treatment and relative humidity were considered fixed factors, with experiment as a random factor and incubation day as a repeated measures factor. The interaction between the fixed factors was also considered in the model. The best covariance structure for the repeated measures data was selected according to the lowest value of the Akaike and Schwarz’s Bayesian information criteria fit statistics [28,29]. The model was fitted using a restricted maximum likelihood estimation method. Differences among treatments were evaluated by LSD range test if statistical significance was found. The significance level was *p* ≤ 0.05. The linear mixed-effects model was performed using SPSS 17.0 statistical program.

## 3. Results

No significant differences were observed between natural brown rot and latent infection incidence recovered on nectarines “Alba Red” and “Red Jim” after 7 d of incubation without postharvest handling (Table 2). Percentage of natural latent infection incidence on nectarines showed a range between 6.7% to 10% before postharvest handling treatments (Table 2). *Monilinia laxa* was the main species isolated on nectarines “Alba Red”, while *M. fructicola* was the main species that was isolated on nectarines “Red Jim” (Table 2).

The latent infection incidence on nectarines incubated at 25 °C for 20 d at both relative humidities showed a progressive increase during the first nine days of the incubation that reached a plateau for the remaining 11 d of the 20 d incubation (Figure 1A). The degree-days and incubation days provided the same curve progress of latent infection in control nectarines at a constant temperature of 25 °C (Figure 1B). Most of the latent infections on control nectarines were activated and/or developed during the first 225 h at 25 °C (Figure 1B). Maximum latent infections were detected in 12.78% of the control nectarines after 20 d of incubation at 25 °C, where the total number of DD was 500 (Table 3).

AULIC without postharvest handling (control nectarines) showed a range between 287.61 in “Red Jim” and 116.67 in “Alba Red” (Figure 2). AULIC variability was divided into contributions of two factors: cultivars (*p* = 0.0063) and treatments (*p* = 0.0005). No significant interaction was observed between both factors (*p* = 0.1323). AULIC in both cultivars for all treatments did not show significant differences between the two different incubation RHs (75% or 100%) (*p*-value for RH from ANOVA = 0.8156 (Appendix A)).

Nectarines stored at 4 °C for several days reduced their total number of DD after 20 d of incubation (Table 3). Thus, maximum total numbers of DD after 10, 11, and 13 d at 4 °C plus 7 d at 25 °C [treatments 7, 8, and 9] were 290.08, 269.08, and 227.08, respectively. Only treatment 2 required the same total number of DD than the control nectarines (Table 3). Nectarine postharvest handling at 4 °C significantly reduced the incidence of latent infections caused by *Monilinia* spp. in both nectarines (Figure 2). Only “Red Jim” nectarines that had been dumped in water at 15 °C [treatment 2] showed the same AULIC value as control nectarines (Figure 2). The highest reduction of AULIC was recorded on both cultivars after 13 d at 4 °C plus 7 d at 25 °C [treatment 9] (Figure 2).

Reduction of latent infection incidence on nectarines was also observed by the linear mixed-effects model after all postharvest handling treatments at 4 °C (Table 4).

The incidence of latent infection in the different treatments was compared using a linear mixed-effects model (Table 4). Treatment and relative humidity were considered fixed factors, with experiment as a random factor and incubation day as a repeated measures factor.

The analysis showed that the incidence of latent infections after cold postharvest treatments was significantly lower than what was observed in nectarines storage at 25 °C for 20 d, regardless of relative humidity (Table 5). The significantly higher mean latent infection incidence corresponded to control treatments that do not include refrigeration [treatments 1 and 2].

Latent infection incidence was fitted to a modified Gompertz model for physiological time description (DD) (Figure 3). The coefficients of determination for the modified Gompertz model ranged from 0.94 to 0.99, indicating the suitability of the function used to fit the latent infection incidence at postharvest conditions (excluding treatment 9 with R^2^ = 0.81) (Table 5). The lag phase to start the activation and/or development of latent infection ranged between 4 to 14 d after postharvest handling treatments at 4 °C (Table 5). Nectarine postharvest handling increased the estimated lag phase of latent infection activation and/or development before the start of the exponential phase (Figure 3). The Gompertz modified model estimated the physiological time to initial activation and/or development of latent infection in the range 27 to 115 DD, except treatment 9 where initial activation and/or development of latent infection estimated was 400.96 (*p* = 0.82139) (Table 5). The activation and/or development of latent infections in the postharvest period reached a stationary phase between 150-225DD (Figure 3). Only treatment 9 was maintained in the exponential phase for the entire incubation time (Figure 3).

The range of rate growth of latent infection predicted by the modified Gompertz model for physiological time description (DD) was approximately 0.047 to 0.573, except in treatment 9 where the rate of growth of latent infection estimated was 19.21 (*p* = 0.97168) (Table 5). The maximum rate growth values were nectarines from treatments 7, 8, and 9 (Table 5). The estimated maximum incidence of latent infections on nectarines ranged from 4.5% [treatment 6] to 12.77% [treatment 8], except treatment 9 (Table 5). The minimum percentage of latent infections predicted by this model at the end of the storage period was observed by treatments 3, 4, 5, 6, and 7 in the range 4.5% [treatment 6] to 7.49% [treatment 5] (Table 5, Figure 3).

## 4. Discussion

The activation and/or development of natural latent infections caused by *Monilinia* spp. in nectarines during the postharvest period was mainly related to temperature and incubation time. In the present work, DD (Day-degree) was used to know the heat accumulation by *Monilinia* spp. to predict latent infection activation and/or development rates such as the date that brown rot latent infection will be visualized. The thermal requirement for the latent infection activation and/or development on harvested nectarines at 25 °C for 20 d of incubation was 500 DD, where a maximum of latent infections (IL = 12.78%) was detected. However, the natural incidence of latent infections caused by *Monilinia* spp. in harvested fruit before their postharvest handling showed a range between 6.7% to 10% after 7 d of incubation at 25 °C (total number of DD was 175). Growing degree days, a similar linear unit system to DD, was widely used to predict crop development with air temperature [30].

The incidence of latent infection and AULIC were reduced by all postharvest treatments, except the immersion of “Red Jim” nectarines in the water dump tank without cooling [treatment 2]. Storing nectarines at 4 °C at intervals during the fruit’s postharvest period extended the latency period of pre-harvest latent infections caused by *Monilinia* spp., regardless of whether this storage occurred before or after fruit disinfection by water dumping. It is widely accepted that the cold storage of harvested fruit is the best method to reduce the growth of several fruit pathogens and to maintain fruit quality [31,32,33]. Lowering the temperature of harvested fruit as quickly as possible (a) decelerates the ripening process by inhibiting enzymatic degradation, suppressing respiratory activity and reducing ethylene production [34,35,36] and (b) reduces the brown rot development [37]. Bernat et al. [37] reported that the decay rate of stone fruit and mycelial development of *M. fructicola* and *M. laxa* increases when the temperature of harvested fruit increases from 0 °C to 25 °C. The longest latency period of latent infections caused by *Monilinia* spp. was preserved in those postharvest treatments with more than 10 d of cold storage. *Monilinia* growth was very low at 0 °C. *M. fructicola* conidia on fruit surface at 0 °C and 100% RH or 4 °C and 100% RH for up to 30 d caused a very low risk of infection, with only a 3.3% and 3.8% brown rot rate of incidence, respectively [26]. *Monilinia* spp. conidia germination still occurs at 0 °C, but they take a longer time to get a very short germ tube and they are unable to produce mycelia [20,38,39].

The lowest DD for the latent infection activation and/or development on postharvest nectarines for 20 d of incubation was recorded in those treatments with more than 10 d of cold storage in the total postharvest period of 20 d. Similar levels of thermal requirement (225DD) were required to reach 10.93% activation and/or development of latent infections caused by *Monilinia* spp. in the first nine days of storage of nectarines at room temperature. These results show the importance of storing nectarines at low temperatures, especially in the early stages of postharvest. Degree-days could be an important component in latent infection simulation models and managerial decisions. Traditional methods for calculating DD assume linear developmental responses to temperature and cannot precisely account for the delay in growth or development at temperatures above the optimal temperature [40]. Physiologically, we have assumed that below a certain base temperature level (0 °C), *Monilinia* spp. growth and development will cease, such as described by Yang et al. [30] in crops. This base temperature should be similar for all developmental stages [30]. The optimum temperature for conidial germination, mycelial growth, sporodochia production, and the highest incidence of brown rot from *M. laxa* and *M. fructicola* is 25 °C [6,37,38,39]. However, no germination of *Monilinia* spp. conidia and rotting on fruit occurred above 38 °C [5,37]. Although the optimal environmental conditions for conidial germination (25 °C and 80 to 100% RH) by *M. fructicola* and *M. laxa* are similar, *M. fructicola* grows faster and sporulates more abundantly than *M. laxa* when the temperature is in the range of 15–25 °C [7,20]. The opposite is true for *M. laxa* when *M. laxa* and *M. fructicola* isolates grow at low temperatures on culture media and fruit tissues [5]. The dissimilar abilities of *M. laxa* and *M. fructicola* to germinate and form appressoria at low temperatures conferred a competitive advantage to *M. laxa* to survive during fruit postharvest refrigeration and cold storage at 4 °C [5]. However, we did not observed any advantage from *M. laxa* over *M. fructicola* on natural latent infection incidence in the present study.

Temperature is a dominant factor controlling *Monilinia* spp. development and their latent infections, although it is also dependent on relative humidity (RH), wetness duration, and water availability [9,21,39,41]. During stone fruit postharvest, humidity must be well controlled and kept constant at 60% in order to avoid new brown rot infections, although at this relative humidity, fruit would lose its firmness and quality, reducing their shelf life [26]. Previously, we reported that high relative humidity levels increase the number of active infections by *M. fructicola* and especially *M. laxa* conidia in in vitro experiments [5]. However, we did not observe any effect of environmental relative humidity on the percentage of incidence of latent infections along the postharvest handling treatments. However, *M. fructicola* mycelia are present in the sub-epidermal tissues in latent infections caused by *Monilinia* spp. when the environmental relative humidity is high [6].

The use of DD has improved our description and prediction of latent infections caused by *Monilinia* spp. on nectarines at postharvest conditions and has allowed us to compare different cold storage approaches. The Gompertz modified model was fitted to the latent infections caused by the *Monilinia* spp. developmental rate data recorded for DD tested and this model described well the relationship between latent infection activation and/or development and temperature at postharvest conditions. This model could be used for predicting the latent infections caused by *Monilinia* spp. at postharvest conditions. The linear model is widely used to describe the relationship between developmental rate and temperature because of its simplicity and easiness to calculate and apply in practice [42,43]. The Gompertz modified model has already been fitted to fungal growth curves in the study of the combined effects of environmental factors such as temperature, pH, water activity, and redox potential [44]. The Gompertz model is very similar to the logistic model but differs in that the turning point is achieved in the first part of the growth cycle and therefore has presented a better predictive adjustment in the analysis of the growth of certain living beings [45]. The maximum activation and/or development of latent infections was reached at 150–225 DD with any postharvest handling less than 11 days at 4 °C, while more cold days maintained the exponential phase of latent infection activation and/or development. Thus, it was interesting to know the shelf life of the fruit, i.e., how long it takes the fruit population to migrate from an asymptomatic status to maximum activation and/or development of latent infection. The estimated growth rate of latent infections in nectarines stored in cold for more than 11 days was much higher when they were kept at room temperature after the cold period was over. Furthermore, the maximum value of latent infections at more than 11 days of cold storage was much higher than that observed in nectarines with less cold days of storage. This unexpected increase could be due to the fact that after more than 11 d at 4 °C and more than 9 d at 25 °C, the physiological state of the fruit changed or suffered chilling injury. The storage of peaches at low temperatures for prolonged periods could induce a form of chilling injury, characterized by breakdown of internal tissues, a lack of juiciness, and a mealy texture [46,47]. Both peaches and nectarines exhibit this disorder, with variation in susceptibility between different cultivars [48].

Surface disinfection is an important practice to avoid secondary infections in the postharvest handling of stone fruit [26]. On the other hand, disinfection of fruit by dumping in water with chlorine will sanitize fresh produce and may reduce decomposition by lowering the effective fungal concentration of conidia [49]. Chlorine, sodium hypochlorite, peracetic acid, and hydrogen peroxide are the most common aqueous disinfectants used in packing houses to disinfect fruit when it arrives from the field and also to clean the surfaces of bins or facilities. Sodium hypochlorite has shown great disinfecting power on surfaces infected with *M. fructicola* [50]. Chlorine at a concentration of 50 mg L^−1^ significantly decreased conidial germination of *Mucor piriformis* and *Penicillium expansum* after 30 s of treatment, although fruit rotting was not controlled. However, in the present study, all nectarines were disinfected with sodium hypochlorite before postharvest treatments, but the incidence of natural latent infections of *Monilinia* spp. was similar to that reported in previous studies (0% to 30%) [10,11,16,17]. Thus, sodium hypochlorite seems to have had only a superficial effect on Monilinia conidia, but not on latent infections below the fruit cuticle.

For harvested stone fruit, the postharvest handling system comprises several operations and stages. The activation and/or development of latent infections caused by *Monilinia* spp. in fruit during the postharvest period was mainly related to temperature and incubation time, but not to relative humidity. The Gompertz model could be used for predicting the activation and/or development of latent infections caused by *Monilinia* spp. at postharvest conditions. Knowing the temperature of each of the postharvest steps and the number of hours the fruit spends at each temperature could make it possible to predict the activation and/or development of pre-harvest latent infections in each fruit batch in order to manage the period of their postharvest handling. Nectarines should be kept cold during the post-harvest period, which will reduce the development of latent infections. However, the number of days in cold storage should not be too great, nor should it exceed the optimum physiological state since if the development of latent infections is exceeded, it will be practically exponential.

## Figures and Tables

**Figure 1 jof-06-00266-f001:**
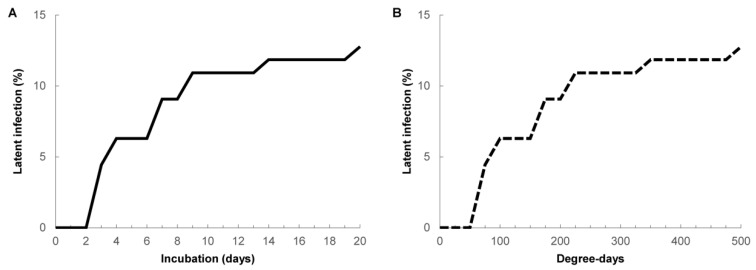
Latent infection incidence on nectarines incubated at 25 °C for 20 d (**A**) or 500 DD (degree-days) (**B**) for overall cultivars in Treatment 1 (without any refrigeration period).

**Figure 2 jof-06-00266-f002:**
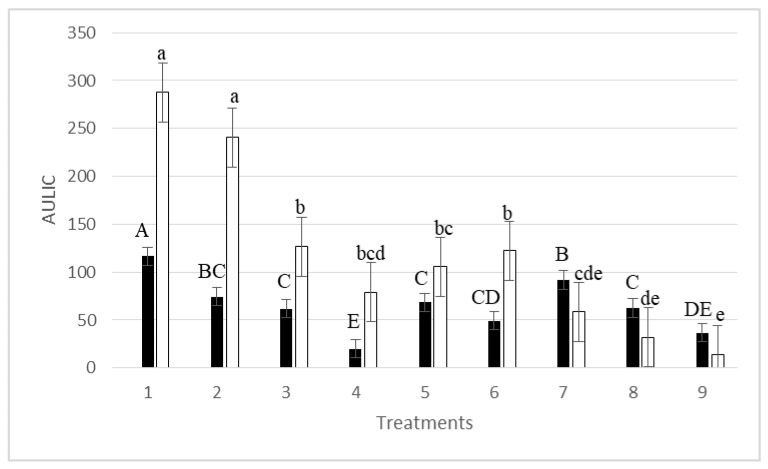
AULIC (area under latent infection curve) on “Red Jim” (□) and “Alba Red” (■) nectarines after postharvest treatments (see Table 1). Columns are the mean of 60 fruit ± standard error. Data were analyzed by a two way-ANOVA (cultivar/treatment). Columns with the same letter in each cultivar are not significantly different (*p* < 0.05) from each other by Student Newman Keuls test. Uppercase and lower case refer to “Red Jim” and “Alba Red” nectarines, respectively.

**Figure 3 jof-06-00266-f003:**
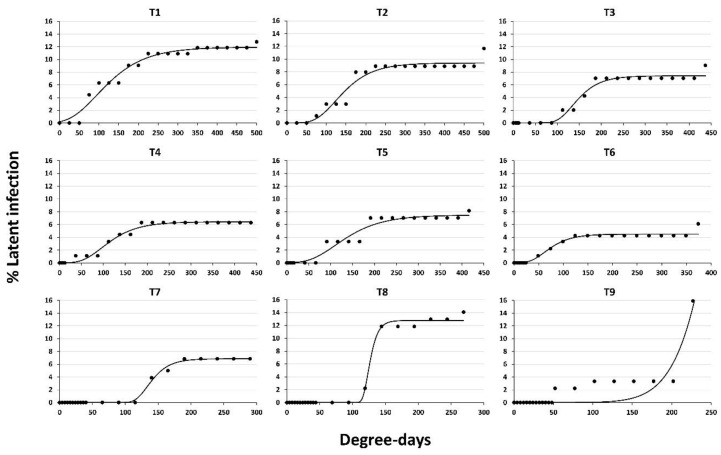
Latent infection incidence of *Monilinia* spp. on both nectarines fitted to the Gompertz model for physiological time description (DD) under different postharvest conditions: (T1) at 20 °C for 20 d; (T2) at 20 °C for 20 d after water dumping for 10 minutes at 15 °C; (T3) cold storage for 3 d at 4 °C after water dumping; (T4) cold storage for 3 d at 4 °C before water dumping; (T5) cold storage for 1 d at 4 °C before water dumping, and cold storage for 3 d at 4 °C after water dumping; (T6) cold storage for 3 d at 4 °C before water dumping, and cold storage for 3 d at 4 °C after water dumping; (T7) cold storage for 10 d at 4 °C after water dumping; (T8) cold storage for 1 d at 4 °C before water dumping, and cold storage for 10 d at 4 °C after water dumping, and (T9) cold storage for 3 d at 4 °C before water dumping, and cold storage for 10 d at 4 °C after water dumping. All treatments were evaluated at either 75% RH or 100% RH. Data were the mean of both. In order to simulate the commercial and market period, the nectarines were incubated at 25 °C under fluorescent lighting (100 μE m^−2^ s^−1^ with a 16 h photoperiod) for at least 7 additional days after treatment until the total incubation period of each treatment was 20 d.

**Table 1 jof-06-00266-t001:** Details of the different postharvest handling treatments. The temperature in pre-cooling and the cold storage chamber was 4 °C at either 75% relative humidity (RH) or 100% RH. The temperature in water dumping was 15 °C for 10 min.

Treatment	Days on Pre-Cooling Chamber	Water Dumping	Days on Cold-Storage Chamber	Total Days at 4 °C	Total Days at 25 °C
1	0	No	0	0	20
2	0	Yes	0	0	20
3	0	Yes	3	3	17
4	3	Yes	0	3	17
5	1	Yes	3	4	16
6	3	Yes	3	6	14
7	0	Yes	10	10	10
8	1	Yes	10	11	9
9	3	Yes	10	13	7

**Table 2 jof-06-00266-t002:** The incidence of brown rot and latent infections caused by *Monilinia* spp. in harvested “Red Jim” and “Alba Red” nectarines after 7 days of incubation at 25 °C under fluorescent lighting (100 μE m^−2^ s^−1^ with a 16 h photoperiod) at 100% RH, and the frequency of *Monilinia* spp., which were isolated from the harvested nectarines before postharvest handling experiments (without any refrigeration period).

Experiment	Nectarine Cultivar	Orchard Location	Brown Rot Incidence (%)	Incidence of *Monilinia* Latent Infection (%)	Frequency of *M. fructicola* Isolates (%)	Frequency of *M. laxa* Isolates (%)
1	Red Jim	Sudanell	26.6a	10.0a	16.7a	1.7b
2	Alba Red	Ivars de Noguera	43.3a	6.7a	3.3b	21.7a
χ^2^			1.172	0.000	4.537	9.784
*p*-value			0.279	1.000	0.033	0.002

The natural incidence of brown rot and latent infections in harvested fruit caused by *Monilinia* spp. in each cultivar before their postharvest handling was determined by placing 30 nectarines or 30 frozen surface-disinfected nectarines, respectively, in a humidity chamber lined with sterilized moist filter paper and being incubated. Values with the same letter in each column are not significantly different from each other according to chi-squared tests with Yates’ correction for 2-by-2 frequency tables (*p* < 0.05).

**Table 3 jof-06-00266-t003:** Degree-days above 0 °C (DD) for each postharvest handling treatment and incubation days.

Incubation (Days)	Postharvest Handling Treatments								
	1	2	3	4	5	6	7	8	9
0	0.00	0.00	0.00	0.00	0.00	0.00	0.00	0.00	0.00
1	25.00	24.93	4.08	4.00	4.00	4.00	4.08	4.00	4.00
2	50.00	49.93	8.08	8.00	8.08	8.00	8.08	8.08	8.00
3	75.00	74.93	12.08	12.00	12.08	12.00	12.08	12.08	12.00
4	100.00	99.93	37.01	36.93	16.08	16.08	16.08	16.08	16.08
5	125.00	124.93	62.01	61.93	41.08	20.08	20.08	20.08	20.08
6	150.00	149.93	87.01	86.93	66.08	24.08	24.08	24.08	24.08
7	175.00	174.93	112.01	111.93	91.08	49.08	28.08	28.08	28.08
8	200.00	199.93	137.01	136.93	116.08	74.08	32.08	32.08	32.08
9	225.00	224.93	162.01	161.93	141.08	99.08	36.08	36.08	36.08
10	250.00	249.93	187.01	186.93	166.08	124.08	40.08	40.08	40.08
11	275.00	274.93	212.01	211.93	191.08	149.08	65.08	44.08	44.08
12	300.00	299.93	237.01	236.93	216.08	174.08	90.08	69.08	48.08
13	325.00	324.93	262.01	261.93	241.08	199.08	115.08	94.08	52.08
14	350.00	349.93	287.01	286.93	266.08	224.08	140.08	119.08	77.08
15	375.00	374.93	312.01	311.93	291.08	249.08	165.08	144.08	102.08
16	400.00	399.93	337.01	336.93	316.08	274.08	190.08	169.08	127.08
17	425.00	424.93	362.01	361.93	341.08	299.08	215.08	194.08	152.08
18	450.00	449.93	387.01	386.93	366.08	324.08	240.08	219.08	177.08
19	475.00	474.93	412.01	411.93	391.08	349.08	265.08	244.08	202.08
20	500.00	499.93	437.01	436.93	416.08	374.08	290.08	269.08	227.08

**Table 4 jof-06-00266-t004:** Mixed Model Analysis of latent infection incidence along time for the different treatments and under two RH conditions.

		Number of Levels	df	F Values	*p*
Fixed Effects	Intercept	1	1	73.649	0.000
	Treatment	9	8	2.213	0.030
	RH	2	1	0.045	0.832
	Treatment × RH	18	8	1.477	0.171
Total		53	144		

**Table 5 jof-06-00266-t005:** Percentage of latent infection incidence from two cultivars at different postharvest handling treatments, coefficient of determination (R^2^), residual sum of squares (RSS), and parameters ± standard error (A is the maximum latent infection reached, µm is the maximum growth rate, and λ is the lag phase duration before the beginning of latent infection growth) estimated for the modified Gompertz function describing the relationship between latent infection and degree-days above 0 °C. C (DD).

Treatments	Mean Latent Infection (%) *	R^2^	RSS	A (ILmax)	μ_m_ (Rate Growth)	λ (Lag Phase)	Lag Phase (Days) *observed* vs. *predicted*
1	8.57 a	0.97	11.8532	11.94 ± 0.31 (*p* = 0.00000) ^#^	0.065 ± 0.008 (*p* = 0.00000)	27.05 ± 10.97 (*p* = 0.02394)	2-0
2	6.45 ab	0.94	15.9857	9.40 ± 0.34 (*p* = 0.00000)	0.068 ± 0.013 (*p* = 0.00005)	72.64 ± 13.85 (*p* = 0.00005)	2-2
3	4.18 bc	0.97	6.7576	7.41 ± 0.23 (*p* = 0.00000)	0.080 ± 0.015 (*p* = 0.00005)	100.48 ± 9.25 (*p* = 0.00000)	7-6
4	4.04 bc	0.98	2.7736	6.42 ± 0.15 (*p* = 0.00000)	0.053 ± 0.006 (*p* = 0.00000)	51.53 ± 7.91 (*p* = 0.00000)	4-4
5	4.04 bc	0.95	9.8497	7.49 ± 0.36 (*p* = 0.00000)	0.047 ± 0.009 (*p* = 0.00004)	51.91 ± 14.88 (*p* = 0.00262)	7-5
6	2.64 bc	0.96	3.1223	4.50 ± 0.14 (*p* = 0.00000)	0.056 ± 0.012 (*p* = 0.00016)	32.40 ± 8.97 (*p* = 0.00199)	7-6
7	2.06 c	0.99	1.0660	6.86 ± 0.12 (*p* = 0.00000)	0.137 ± 0.015 (*p* = 0.00000)	115.20 ± 3.02 (*p* = 0.00000)	14-13
8	3.70 bc	0.99	3.3767	12.77 ± 0.20 (*p* = 0.00000)	0.573 ± 0.091 (*p* = 0.00001)	115.43 ± 1.18 (*p* = 0.00000)	14-14
9	1.76 c	0.81	47.4394	10,349.6 ± 333,500.5 (*p* = 0.97558)	19.21 ± 533.57 (*p* = 0.97168)	400.96 ± 1750.27 (*p* = 0.82139)	13-16

* Data are the mean of 3–6 replicates, with 10 fruit per replicate. Means with the same letter in each column are not significantly different (*p* ≤ 0.030) after the linear mixed-effects model analysis. Treatment and relative humidity were considered fixed factors, with experiment as a random factor and degree-days above 0 °C as a repeated measures factor. The interaction between the fixed factors was also considered in the model. Differences among treatments were evaluated by the LSD multiple range test (*p* ≤ 0.05). ^#^
*p*-values for the parameter estimates.

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
