# Peer review of "Impact of Postharvest Handling on Preharvest Latent Infections Caused by Monilinia spp. in Nectarines"

_jof, 2020, doi:10.3390/jof6040266_

Round 1

Reviewer 1 Report

The paper shows the modelization of infection on nectarines based on different cooling treatments in order to determine the optimal conditions to minimize infection. The authors collected a great deal of data and perform a fitting mathematical model. I would like to point out that freezing the nectarines to boost the infection, might make the data differ from real market conditions but it is a common practice to obtain consistent infection data. Therefore, this does not devalue the obtained data and it should be published. This being said, I would like to point out some minor comments:

  1. In line 226 add the “s”: “between both factors (P=0.1323).”
  2. In line 242 it is noted that “The highest reduction of AULIC was recorded on both cultivars after 13 d at 4 ºC plus 7 d at 25 ºC [treatment 9] (Figure 2)”. However, treatment 4 seems to have a lower AULIC for Alba Red. If this is true, elaborate on the idea of using different treatments depending on the cultivar.
  3. In legend from figure 3 it is mention that “All treatments were evaluated at either 75 % RH or 100 % RH” but the graphs show no mention to whether the data presented is for 75% or 100%. Indicate if the data in the graphs are for 75, 100% or a mean of both.
  4. The headers of some supplementary tables should be translated to English.
  5. Treatment 1 is referred as “C” in supplementary Material. Please, change “C” to “1” to avoid confusion.

Reviewer 2 Report

This study is aimed to investigate the impact of postharvest handling on preharvest latent infections caused by Monilinia spp. in nectarines. The study is a good practical work but it is not suitable for publication in Journal fo Fungi. This study may be suitable for an applied science journal.

Major concerns

1) Nature and major characteristics of the latent infection of Monilinia spp were published in several publications under both preharvest and postharvest conditions. This study gives no new fundamental scientific facts to the previously published materials. In this ms the major fact is that mature fruit is the most susceptible to Monilinia spp but not all infections are expressed at harvest, consequently, the latent infections will evidently express at postharvest, and of course, the expression level of the symptoms is dependent on the temperature, wetness and postharvest handling. This is a good practical testing but it is not enough for a journal publising basic science on fungi such as Journal of Fungi.

2) Introduction does not highlight the new basic science elements of the work. What are the new basic science aspects of the work? The tested postharvest practical treatments have little basic science aspect.

3) Authors use a model to fit observed data and show temporal dynamics of the obtained results and compare practical treatments. However, no description is given what is the scientific pupose to use the model. Authors state that they will use modified Gompertz model, but no data are given why this model was selected. No indication is given whether the authors have tested any other models as in T9, the model is not really good. Apart from the Gompertz model AULIC data are also given but none of them can add any new scientific values to the manuscript without giving the scientific purpose of the use of both AULIC and the model in order to gain more fundamental knowledge on the investigated fungi. At the present stage, both the AULIC and the model can only mechanically interpret the data on the tested practical treatments. The manuscipt seems to miss the basic science elements.

4) This work contains 3 figures and 5 tables, but most of them are redundant or can be combined. The study has 3 figures and 1 table with suitable amount of results (Figs 1, 2, 3, and Table 3). See details below. In addition to the above points of 1-3, these amounts of results are not robust enough for publication in Journal of Fungi.

Other comments

1) Materials and Methods.

Why do you need a model to fit to your observed data? At this stage it it is only for treatment comparison. You should use your model for gaining fundamental results or for predicting something?

R2 is not suitable for evaluate the goodness-of-fit in case of nonlinear models such as modified Gompertz model. Other statistical measures are needed to evalaute the goodness-of-fit. Authorts should have been tested other nonlinear models. Not sure Gompertz will be the best model for all treatments. It can bee seen that not the same model will fit the best to the data of all treatments. This is especially true for T9.

2) Most Tables are redundant or can be combined with Figures.

Table 2: Contains basically 6 results data. Therefore this table is not needed. This data can be included in the text and the table can be deleted.

Table 4. RH or HR? The relevenat info of this statsitical table (the 4 P values) can be included in the text. This table is not needed it can be deleted.

Table 5 includes the components of the model, which can be included in Figure 3 as equations for each of the treatments. Therefore, Table 5 is not needed and the table can be deleted.

3) In addition, the study is negligently written, and some basic roles were not followed during the preparation of the ms. Some examples

Title position of tables: title of Table 1 is under the table and for other tables it is above the table.

The names of the cultivars are inconsequently used: e.g. cv. Red Jim, ‘Red Jim’ or Red Jim. cv. was never introduced.

Beginning of the sentence should not start with abbrevaitions: e.g. DD (L156), M. laxa (L195)

The Figures and Tables are not self-explanatory

- Figure 1: Title: on which treatments ? on which cultivars? Or this is for overall treatments and overall cultivars?

- Figure 2: What do you mean bars? Where are the bars? You have columns but not bars.

- Figure 3: Title: on which cultivars? on which fungal species? Dimension of Y axis is not in the right place. Quality of this figure is bad. Numbers can not be seen.

- Table 3: This is a row data set without any analyses. Title is not self explanatory. No indications are given for 1-9. on which cultivars? on which fungal species?

ºC and not underlined ºC

’et al.’ or ’et al’

and so on …

4) References

You should have been considered mdpi format: some examples:

L496: Modeling of the bacterial growth curve.

Some examples: journal name: Pl. or Plant (L529)

Journal name italic or not italic (L544)

DOI is sometimes provided and sometimes not provided

Reviewer 3 Report

Please find attached a pdf document with the comments on your paper
